# Effects of Vitamin D_3_ and 25(OH)D_3_ Supplementation on Growth Performance, Bone Parameters and Gut Microbiota of Broiler Chickens

**DOI:** 10.3390/ani15192900

**Published:** 2025-10-04

**Authors:** Rakchanok Phutthaphol, Chaiyapoom Bunchasak, Wiriya Loongyai, Choawit Rakangthong

**Affiliations:** Department of Animal Science, Faculty of Agriculture, Kasetsart University, Bangkok 10900, Thailand; rakchanok.ph@ku.th (R.P.); agrchb@ku.ac.th (C.B.); agrwyi@ku.ac.th (W.L.)

**Keywords:** vitamin D_3_, 25(OH)D_3_, growth performance, tibia, broiler, cecal microbiota

## Abstract

**Simple Summary:**

Broiler chickens grow quickly but are usually raised indoors without access to sunlight, which limits their ability to produce vitamin D naturally. Vitamin D is essential for bone strength, growth, and overall health. A special form of vitamin D, called 25-hydroxycholecalciferol [25(OH)D_3_], is more easily used by the body than normal vitamin D_3_. In this study, we tested whether adding 25(OH)D_3_ to diets could benefit broilers. A total of 952 birds were fed diets with no vitamin D_3_, standard vitamin D_3_, or vitamin D_3_ plus two levels of 25(OH)D_3_. Birds fed diets with either vitamin D_3_ or 25(OH)D_3_ grew faster and used feed more efficiently than birds without vitamin D_3_, while growth was similar between the vitamin D_3_ and 25(OH)D_3_ groups. Bone strength did not differ significantly among treatments, although birds given the higher level of 25(OH)D_3_ showed a slight numerical trend toward stronger bones. We also studied gut bacteria and found that 25(OH)D_3_ changed the balance of microbes: it reduced excess *Lactobacillus* and increased bacteria linked with fiber digestion and nutrient absorption. These results show that 25(OH)D_3_ is more effective than vitamin D_3_ alone, helping broilers grow better, use feed more efficiently, and maintain a healthier gut.

**Abstract:**

Broiler chickens are commonly reared in closed housing systems with limited exposure to sunlight, thereby relying entirely on dietary sources of vitamin D. The hydroxylated metabolite 25-hydroxycholecalciferol [25(OH)D_3_] has been proposed as a more potent form than native vitamin D_3_ (cholecalciferol). This study evaluated the effects of dietary supplementation with vitamin D_3_ alone or in combination with 25(OH)D_3_ on growth performance, bone characteristics, and cecal microbiota in Ross 308 broilers. A total of 952 one-day-old male chicks were allocated to four treatments: a negative control (no vitamin D_3_), a positive control (vitamin D_3_ according to Ross 308 specifications), and a positive control supplemented with 25(OH)D_3_ at 1394 or 2788 IU/kg, in a randomized design with 17 replicates per treatment and 14 birds per replicate. Over a 40-day feeding trial, diets containing vitamin D_3_ (positive control) or supplemented with 25(OH)D_3_ significantly improved final body weight, weight gain, average daily gain, and feed conversion ratio compared with the negative control (*p* < 0.01), with no significant differences among the positive control and 25(OH)D_3_-supplemented groups, with a clear linear dose-dependent response. Although tibia ash and bone-breaking strength were not significantly affected, linear responses indicated a slight numerical trend toward improved skeletal mineralization with increasing 25(OH)D_3_. Microbiota analysis indicated that 25(OH)D_3_ affected cecal microbial ecology: low-dose inclusion showed reduced species richness and evenness, whereas high-dose inclusion restored richness to levels comparable to the positive control and enriched taxa associated with fiber fermentation and bile acid metabolism while reducing *Lactobacillus* dominance. In conclusion, supplementation with 25(OH)D_3_ in addition to vitamin D_3_ enhanced growth performance and selectively shaped the cecal microbiota of broilers, with suggestive benefits for bone mineralization. These findings highlight 25(OH)D_3_ as a more potent source of vitamin D than cholecalciferol alone and support its practical use in modern broiler nutrition to improve efficiency, skeletal health, and microbial balance.

## 1. Introduction

Modern broiler production has achieved remarkable improvements in growth rate and feed efficiency through intensive genetic selection. However, the rapid increase in body weight often surpasses skeletal development, predisposing birds to leg disorders and locomotion problems. These conditions compromise animal welfare, restrict feed intake, reduce growth, and ultimately lead to economic losses during processing [1,2].

Vitamin D plays an essential role in calcium and phosphorus absorption, which are critical for bone mineralization [3]. Under natural conditions, vitamin D_3_ (cholecalciferol) is synthesized in the skin upon exposure to ultraviolet B (UV-B) sunlight. Broilers reared in closed housing systems with artificial lighting rarely receive sunlight; thus, their diets must supply adequate vitamin D_3_ [4]. Once ingested, vitamin D_3_ is hydroxylated in the liver to form 25-hydroxycholecalciferol (25(OH)D_3_), the primary circulating metabolite, and subsequently converted in the kidneys into the biologically active 1,25-dihydroxycholecalciferol (1,25(OH)_2_D_3_), which regulates calcium and phosphorus homeostasis and bone health [5,6].

In recent years, 25(OH)D_3_ has become commercially available as a direct dietary supplement in poultry nutrition. Because it bypasses the hepatic conversion step, it may be more efficiently utilized by the body [7,8]. Several studies have reported that 25(OH)D_3_ exhibits higher bioavailability than vitamin D_3_ and can improve growth, bone strength, and reduce leg abnormalities in broilers [9,10].

Beyond skeletal functions, vitamin D also contributes to intestinal health by strengthening the gut barrier, modulating inflammation, and including effects on beneficial taxa such as *Lactobacillus* and suppression of pathogens such as *Salmonella* [11,12]. Although the positive effects of vitamin D_3_ or 25(OH)D_3_ supplementation have been widely documented, limited information is available regarding the potential benefits of combining these two sources at different dietary inclusion levels. Clarifying how such combinations affect growth performance, skeletal development, and gut microbiota could help optimize vitamin D supplementation strategies for modern broiler production. Therefore, the objective of this study was to evaluate the effects of dietary vitamin D_3_ alone and in combination with two graded levels of 25(OH)D_3_ on growth performance, tibial bone characteristics, and cecal microbiota composition in broiler chickens.

## 2. Materials and Methods

### 2.1. Ethical Approval and Animal Care

All experimental procedures were reviewed and approved by the Institutional Animal Care and Use Committee of Kasetsart University, Bangkok, Thailand (ACKU68-AGR-025). Birds were managed in accordance with the guidelines for the care and use of agricultural animals in research and teaching [13].

### 2.2. Animals, Experimental Design, and Husbandry

The trial was conducted at Sun Broiler Farm, Nakhon Nayok Province, Thailand. A total of 952 one-day-old male Ross 308 broiler chicks (initial body weight: 41.5 ± 0.5 g) were used in a 40-day feeding trial. Upon arrival, birds were randomly assigned to four dietary treatments using a completely randomized design (CRD), with 14 replicate pens per treatment and 17 birds per pen. Birds were housed in an environmentally controlled facility equipped with an evaporative cooling system. Each of the 56 pens (1 × 2 m; 2 m^2^ floor area) was bedded with 5 cm-deep rice husk litter. Feed and water were provided ad libitum throughout the experiment. Lighting was maintained at 50 lux according to a standard broiler lighting program (18 h light: 6 h dark), in compliance with the Thai Agricultural Standard TAS 6901-2017 [14]. All birds were vaccinated according to commercial practices: Newcastle disease (day 1), infectious bursal disease (days 1 and 20), and infectious bronchitis (days 1 and 17).

### 2.3. Experimental Diets

Experimental diets were formulated with maize, wheat, and soybean meal, and nutrient composition was calculated based on Ross 308 nutritional recommendations [15]. Four dietary treatments were tested:-Negative control (NC): Negative control (NC): Diet without vitamin D_3_ supplementation in the premix, used as a biological baseline rather than a commercial model, with welfare carefully maintained.-Positive control (PC): Diet supplemented with vitamin D_3_ according to Ross 308 recommendations.-PC + 25(OH)D_3__1X: Diet supplemented with vitamin D_3_ and 25(OH)D_3_ (commercial product: Bio D^®^, supported by Huvepharma, Bangkok, Thailand) was supplemented at levels of 1394 IU/kg diet.-PC + 25(OH)D_3__2X: Diet supplemented with vitamin D_3_ and 25(OH)D_3_ from the same source at 2788 IU/kg.

All diets were provided in pellet form and were isoenergetic and isonitrogenous. The detailed ingredient composition and nutrient values are presented in Table 1, and the treatment allocations are summarized in Table 2.

### 2.4. Growth Performance

Growth performance was evaluated across three growth phases: days 1–11 (starter), days 12–30 (grower), and days 31–40 (finisher). Body weight and feed intake of each replicate pen were recorded at the end of each period. From these data, body weight gain (BWG), average daily gain (ADG), feed conversion ratio (FCR), and mortality rate were calculated. For mortality adjustment, the body weight of dead birds and the amount of residual feed in the corresponding pen were recorded immediately to correct performance calculations. Flock uniformity was assessed in each phase by calculating the coefficient of variation (CV, %) based on the standard deviation of body weight within each pen.

### 2.5. Bone Parameters

At 40 days of age, sixty birds (15 per treatment) with body weights closest to the pen average were selected for bone sampling. The left tibia was collected for ash content analysis, and the right tibia was collected for mechanical strength measurement. Soft tissues were removed, and bones were sealed in labeled plastic bags and stored at −20 °C until analysis.

#### 2.5.1. Tibia Ash Analysis

Left tibiae were thawed, dried, and finely ground. Ash content was determined following AOAC procedures [16]. Approximately 5–10 g of bone powder was weighed into pre-weighed porcelain crucibles. Samples were charred on a hot plate under a fume hood until smoke emission ceased and the residue turned black. Crucibles were then placed in a muffle furnace at 550 °C for 3 h. If necessary, further incineration was performed until consistent white or gray ash was obtained. After cooling in a desiccator for 30 min, crucibles were reweighed, and tibia ash content was expressed on a dry matter basis.

#### 2.5.2. Tibia Breaking Strength

Right tibiae were used for mechanical testing. After thawing, bones were positioned horizontally on a three-point bending device, with the midpoint subjected to vertical compression using a texture analyzer. Compression was applied at a rate corresponding to 3–6% of the expected ultimate load, with a force increase of 0.9–1.2 MPa/min (9–12 kgf/cm^2^/min), until fracture occurred. The maximum force at the breaking point was recorded as tibia breaking strength (N).

### 2.6. Gut Microbiota Analysis

#### 2.6.1. Sample Collection

At 40 days of age, a total of 28 birds (n = 7 replicates per group) with body weights closest to the treatment mean were selected for cecal sampling. Birds were euthanized by carbon dioxide (CO_2_) stunning. Cecal were excised, ligated with sterile string, and immediately placed into sterile test tubes. Samples were sealed, stored at −20 °C, and subsequently used for nucleic acid extraction to preserve microbial community composition.

#### 2.6.2. DNA Extraction and 16S rRNA Gene Amplicon Sequencing

Microbial DNA was extracted from 300 mg of cecal content per sample using the DNeasy PowerFecal Pro DNA Kit (Qiagen, Germantown, MD, USA). The V3–V4 region of the 16S rRNA gene was amplified with primers 341F (CCTACGGGNGGCWGCAG) and 805R (GACTACHVGGGTATCTAATCC) using 2X sparQ HiFi PCR Master Mix (QuantaBio, Beverly, MA, USA). PCR conditions were: 98 °C for 2 min; 30 cycles of 98 °C for 20 s, 60 °C for 30 s, and 72 °C for 1 min; and a final extension at 72 °C for 1 min. Amplicons were purified with sparQ Puremag Beads, indexed with Nextera XT primers (5 µL each, 8–10 cycles), pooled, and diluted to 4 pM. Sequencing was performed as 250 bp paired-end reads on an Illumina MiSeq platform (Omics Sciences and Bioinformatics Center, Chulalongkorn University, Bangkok, Thailand) [17]. Following quality filtering, sequencing depth ranged from 27,731 to 59,983 reads per sample, with an average of 41,407 high-quality reads.

#### 2.6.3. Bioinformatics and Statistical Analysis

Raw sequencing reads were processed using Quantitative Insights into Microbial Ecology 2 (QIIME 2, version 2020.8) [18]. Sequences were adapter-trimmed, demultiplexed, and low-quality reads and chimeras were removed using the DADA2 plugin. High-quality reads were clustered into operational taxonomic units (OTUs) at 99% similarity against the SILVA 16S rRNA reference database (version 132). Taxonomic classification was assigned using a naïve Bayes classifier trained on the SILVA database. Rarefaction was performed at a depth of 27,731 sequences per sample. Alpha diversity (Observed OUTs, Shannon, Pielou’s evenness, and Faith’s PD) were compared by Kruskal–Wallis with Benjamini–Hochberg correction. Beta diversity was analyzed based on unweighted and weighted UniFrac distances, visualized with principal coordinate analysis (PCoA), and tested for significance using PERMANOVA (999 permutations).

### 2.7. Statistical Analysis

Data on growth performance and bone parameters were analyzed using the General Linear Model (GLM) procedure in SAS software (version 9.0) based on a completely randomized design with pen as the experimental unit. Model assumptions were verified (Shapiro–Wilk normality; Levene’s test for homogeneity). Results are presented as means with pooled standard error of the mean (SEM). Differences among treatment means were assessed using Tukey’s HSD test [19]. Orthogonal polynomial contrasts were applied to test linear and quadratic responses to increasing dietary vitamin D supplementation. Statistical significance was declared at *p* < 0.05, and highly significant differences were reported at *p* < 0.01.

## 3. Results and Discussion

### 3.1. Growth Performance

Supplementation with 25(OH)D_3_, in addition to standard dietary vitamin D_3_, significantly improved broiler growth performance throughout the 40-day trial (Table 3). Birds receiving 25(OH)D_3_ at 1394 and 2788 IU/kg exhibited higher final body weight (BW), cumulative body weight gain (BWG), average daily gain (ADG), and improved feed conversion ratio (FCR) compared with the negative control (no vitamin D_3_) (*p* < 0.01). However, no significant differences were observed between the positive control (vitamin D_3_ only) and the 25(OH)D_3_-supplemented groups. These improvements followed a dose-dependent linear trend (*p* < 0.01), whereas no quadratic responses were detected. In contrast, feed intake (FI) and mortality rates were unaffected (*p* > 0.05).

The positive effects of 25(OH)D_3_ were most evident during the starter (0–11 days) and grower (12–30 days) phases. In the starter phase, birds supplemented with 25(OH)D_3_ showed significantly higher BW, BWG, and ADG compared with controls, with a clear linear dose–response (*p* < 0.01). Although FI was not affected, FCR exhibited a mild linear improvement (*p* = 0.02), suggesting early enhancements in feed efficiency. These early-phase benefits likely reflect the superior biological potency of 25(OH)D_3_ relative to cholecalciferol [9]. Unlike vitamin D_3_, 25(OH)D_3_ bypasses hepatic hydroxylation and is more rapidly converted into the active metabolite 1,25-dihydroxycholecalciferol [1,25(OH)_2_D_3_], which enhances calcium and phosphorus absorption through the upregulation of intestinal transport proteins such as TRPV5-6, calbindin-D9k, and PMCA1b [20,21]. Enhanced mineral absorption during the post-hatch period is crucial for skeletal development, thereby supporting efficient muscle accretion and overall growth [9,22].

In the finisher phase (31–40 days), the benefits of vitamin D supplementation were less pronounced. Final BW was significantly higher in supplemented groups (*p* < 0.01), whereas BWG, ADG, FI, and FCR showed only numerical, non-significant improvements, and mortality was unaffected. This attenuation likely reflects age-related shifts in growth dynamics, as skeletal mineralization slows and energy is directed toward maintenance and fat deposition [10]. Nevertheless, the numerically lower FCR in supplemented groups aligns with previous reports that vitamin D metabolites can modestly enhance feed efficiency through sustained effects on gut integrity and nutrient utilization [9,23].

Over the entire 40-day period, cumulative BW, BWG, ADG, and FCR were significantly improved in birds receiving 25(OH)D_3_, particularly at 2788 IU/kg (*p* < 0.01). The absence of adverse effects on feed intake or mortality suggests that supplementation at these levels is both efficacious and safe. Mortality rates did not differ among treatments and were within the expected range for tropical broiler production without antibiotic growth promoters, indicating that the diets did not compromise survival or welfare. Collectively, these findings confirm the dose-dependent efficacy of 25(OH)D_3_ in enhancing broiler performance through improved mineral metabolism and feed efficiency [23]. Moreover, a recent study in Chinese yellow-feathered broilers reported that dietary vitamin D_3_ supplementation modulated gut microbiota and improved intestinal morphology, which may have contributed to enhanced nutrient absorption and growth [24].

### 3.2. Bone Mineralization and Strength

The effects of dietary vitamin D supplementation on tibia ash percentage and bone-breaking strength of broiler chickens are presented in Table 4. No statistically significant differences (*p* > 0.05) were observed among treatments for tibia ash or breaking strength, either by ANOVA or orthogonal contrast analysis. However, a slight linear trend was noted, with tibia ash (*p* = 0.08) and bone-breaking strength (*p* = 0.06) showing numerical increases in response to higher dietary 25(OH)D_3_ levels.

Broilers receiving the highest 25(OH)D_3_ supplementation (2788 IU/kg) exhibited the highest numerical values for both tibia ash (42.52 ± 1.51%) and bone strength (35.86 ± 7.04 N), indicating only a numerical linear trend rather than a statistically significant improvement. These trends may reflect the influence of 25(OH)D_3_ on bone-specific processes such as osteoblast differentiation, matrix mineralization, and regulation of local endocrine pathways within the growth plate. Previous studies have shown that 1,25-dihydroxycholecalciferol, the active form of vitamin D, not only supports mineral absorption but also modulates bone remodeling by stimulating osteocalcin synthesis and enhancing the deposition of calcium and phosphate into the bone matrix [25,26]. Moreover, vitamin D metabolites play an important role in regulating endochondral ossification during early skeletal growth, a critical process for structural integrity in chickens [27,28].

Although this study assessed bone parameters only at market age (day 40), the observed numerical trends suggest that earlier supplementation with 25(OH)D_3_ may have influenced bone architecture and contributed to higher mineral content and strength at the endpoint. This interpretation is consistent with recent findings indicating that early metabolic programming of bone tissue by vitamin D metabolites can have lasting effects on skeletal quality [29]. While statistical significance was not achieved, the consistent dose-responsive trends observed, coupled with plausible biological mechanisms involving osteoblastic activity and mineral matrix deposition, highlight the potential of 25(OH)D_3_ to positively influence skeletal development in poultry [6,30]. These findings suggest that the skeletal response to 25(OH)D_3_ may be cumulative and time-dependent and may also intersect with other physiological systems. In this context, growing evidence has pointed to the gut microbiota as a critical mediator of nutrient metabolism, bone homeostasis, and immune function.

### 3.3. Microbiota Structure in the Cecal Contents

Alpha diversity (Figure 1) differed among treatments for species richness (Observed OTUs; Kruskal–Wallis, *p* = 0.028). Post hoc testing showed lower richness in the 25(OH)D_3_ 1394 IU/kg group than the positive group (*p* < 0.05) and higher richness in the 25(OH)D_3_ 2788 IU/kg group than the 25(OH)D_3_ 1394 IU/kg group (*p* < 0.05). No overall difference was detected for Shannon diversity (*p* = 0.084), although pairwise comparison suggested that the negative control tended to have a higher Shannon index than the 25(OH)D_3_ 1394 IU/kg group (*p* = 0.025). Pielou’s evenness differed among treatments (*p* = 0.042), with the negative group showing higher evenness than both 25(OH)D_3_ groups (1394 IU/kg: *p* = 0.012; 2788 IU/kg: *p* = 0.025). Faith’s phylogenetic diversity did not differ (*p* = 0.224). Beta diversity (Figure 2) based on PCoA showed no distinct clustering by treatment, and PERMANOVA confirmed non-significant differences for both unweighted UniFrac (*p* = 0.279) and weighted UniFrac (*p* = 0.637), indicating that supplementation affected within-community diversity metrics without overtly restructuring overall community composition.

At the family/genus levels (Figure 3a,b), the negative group was characterized by higher relative abundance of *Lactobacillaceae* and its representative genus *Lactobacillus* than all supplemented groups. Vitamin D_3_ supplementation (positive) and diets containing 25(OH)D_3_ were associated with increased representation of families linked to fiber and bile-acid metabolism (e.g., *Oscillospiraceae*, *Christensenellaceae*), and with shifts in several genera. Specifically, the 25(OH)D_3_ 2788 IU/kg group showed higher proportions of *Alistipes* and enrichment of fiber- and mucus-degrading taxa (*Ruminococcus torques* group, *Papillibacter*, *Merdibacter*), while the 25(OH)D_3_ 1394 IU/kg group displayed relatively higher *Barnesiella* and *Subdoligranulum*; supplemented groups also tended to have more *Clostridia UCG-014*, *Clostridia vadinBB60* group, and *Christensenellaceae R-7* group. Consistent with these patterns, LEfSe (LDA cutoff > 3.0) identified discriminative taxa (Figure 3c): the negative group was enriched in *Lactobacillus* together with higher taxa (*Lactobacillaceae*, *Lactobacillales*), plus members of *Monoglobaceae*, *Defluviitaleaceae*, and *Desulfovibrionales*; the positive group was marked by *Oscillospiraceae* (and other Actinobacteria-related features); and the 25(OH)D_3_ 2788 IU/kg group was marked by *Ruminococcus torques* group, *Papillibacter*, and *Merdibacter*.

The present findings indicate that dietary supplementation with vitamin D_3_ and 25(OH)D_3_ modulated cecal microbial ecology in a dose-dependent manner. Alpha diversity measures revealed that moderate inclusion of 25(OH)D_3_ (1394 IU/kg) reduced species richness and evenness, whereas higher inclusion (2788 IU/kg) restored richness to levels comparable to the positive group. These results suggest that 25(OH)D_3_ exerts selective pressures on microbial communities, potentially through modulation of intestinal epithelial function, antimicrobial peptide secretion, and mucosal immunity mediated by the vitamin D receptor (VDR) [6,31,32,33]. Importantly, although certain taxa showed localized changes in richness and evenness, beta diversity analyses confirmed that the overall community structure remained stable across treatments. This indicates that the effects of vitamin D_3_ and 25(OH)D_3_ represent localized adjustments within the microbial community rather than major restructuring.

Our taxonomic profiling showed that *Lactobacillus* was most abundant in the negative control group, while vitamin D_3_ and 25(OH)D_3_ supplementation reduced its dominance and enriched other functional taxa. Recent evidence indicates that vitamin D influences gut microbiota mainly through VDR signaling [34]. Impaired VDR expression or activation has been associated with reduced *Lactobacillus* populations and expansion of *Proteobacteria*, leading to a more pro-inflammatory environment [35]. Conversely, adequate vitamin D status or butyrate-mediated enhancement of VDR supports microbial balance and intestinal immune function [36]. Thus, the reduction in *Lactobacillus* observed here should be interpreted in context: while lower dominance may support greater diversity, *Lactobacillus* itself also contributes to host health, and its role likely depends on maintaining an appropriate balance within the community.

Supplementation with vitamin D_3_, and particularly 25(OH)D_3_, enriched several bacterial taxa associated with gut homeostasis and immunomodulatory functions. Notably, this included *Oscillospiraceae*, *Ruminococcus* torques group, and *Papillibacter*—genera reported in previous studies as important short-chain fatty acid (SCFA) producers and modulators of immune responses [37,38,39]. Such functions are generally linked with nutrient absorption, intestinal stability, and host metabolic health. Interestingly, the enrichment patterns differed between treatments: *Oscillospiraceae* was identified as a key biomarker in the vitamin D_3_ group, while the *Ruminococcus* torques group and *Papillibacter* were particularly elevated in the high-dose 25(OH)D_3_ group. Collectively, these microbial shifts are thought to enhance metabolic and immunomodulatory potential [40,41,42]. However, as our study did not directly measure SCFA concentrations, intestinal inflammation, or immune responses, these mechanisms remain uncertain and require confirmation in future studies.

## 4. Conclusions

This study demonstrated that dietary supplementation with 25-hydroxycholecalciferol [25(OH)D_3_], in addition to standard vitamin D_3_, improved growth performance of broiler chickens in a dose-dependent manner. Broilers fed diets with vitamin D_3_ or 25(OH)D_3_ showed improved growth performance compared with the negative control, while no significant differences were detected between the vitamin D_3_ and 25(OH)D_3_ groups. Linear responses suggested that potential benefits of 25(OH)D_3_ were more apparent during the early growth phase. Although tibia ash and bone-breaking strength were not significantly affected, linear response patterns indicated only numerical trends toward higher mineralization and skeletal integrity at greater supplementation levels.

Cecal microbiota analysis revealed that vitamin D_3_ and 25(OH)D_3_ selectively modulated microbial diversity and composition. While moderate inclusion reduced species richness and evenness, higher supplementation restored microbial richness and enriched taxa associated with fiber fermentation, bile acid metabolism, and short-chain fatty acid production. These microbial shifts, together with enhanced mineral utilization, likely contributed to the improvements in growth performance and skeletal development observed in this study.

Overall, the findings indicate that 25(OH)D_3_ is a more potent source of vitamin D than cholecalciferol alone, supporting growth efficiency, skeletal health, and microbial balance in broilers. From a practical standpoint, incorporating 25(OH)D_3_ into commercial diets—particularly at higher inclusion levels—may improve feed efficiency and bird welfare. Future research should refine optimal dosages and evaluate long-term impacts on immunity, carcass quality, and resilience under diverse production systems.

## Figures and Tables

**Figure 1 animals-15-02900-f001:**
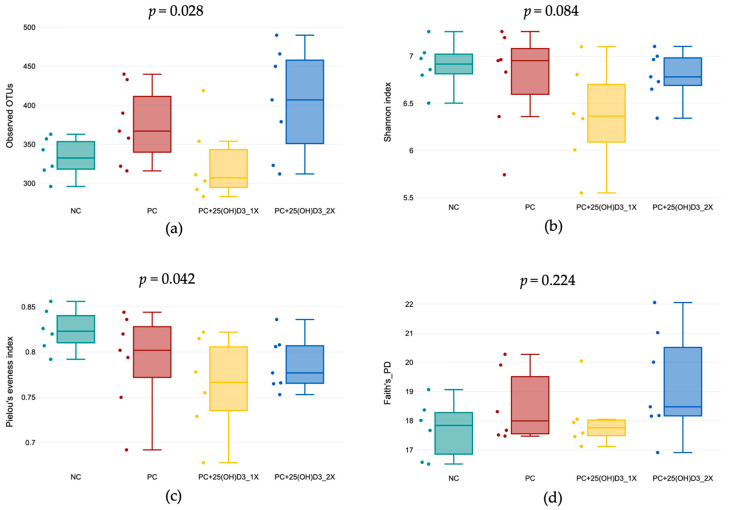
Alpha diversity of cecal microbiota in broilers. (**a**) Observed OTUs, (**b**) Shannon index, (**c**) Pielou’s evenness index, and (**d**) Faith’s phylogenetic diversity (PD) index.

**Figure 2 animals-15-02900-f002:**
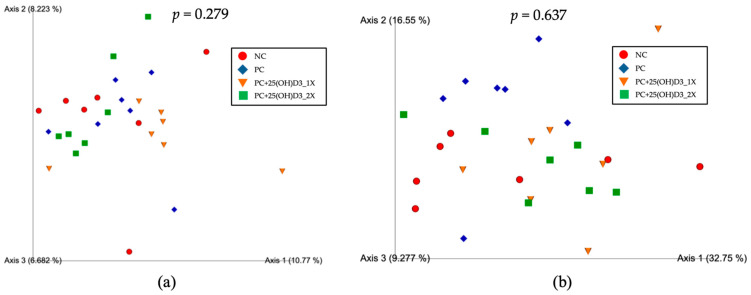
The PCoA Emperor plots from (**a**) unweighted and (**b**) weighted UniFrac distances analyses of cecal microbiota in broilers.

**Figure 3 animals-15-02900-f003:**
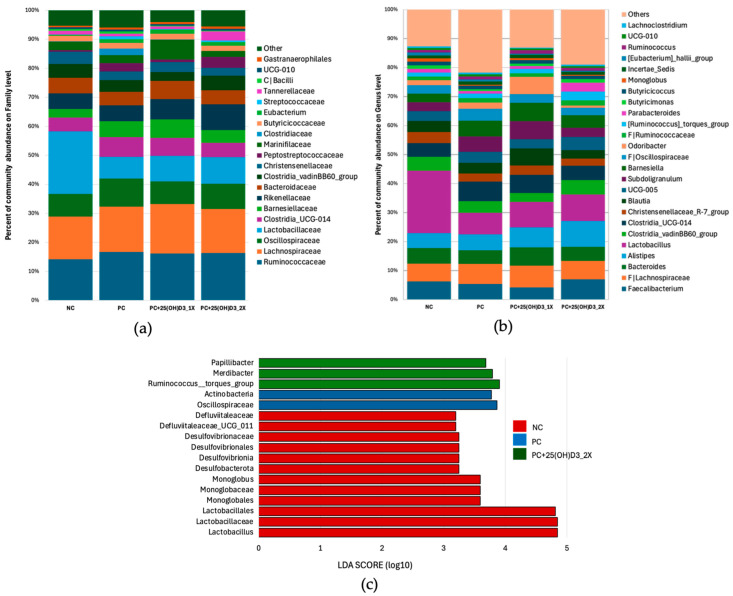
Relative abundance of cecal microbiota in broilers: (**a**) family level, (**b**) genus level, and (**c**) LDA scores of taxonomic biomarkers identified by LEfSe (log_10_).

**Table 1 animals-15-02900-t001:** The composition and the nutrient levels of the basal diets.

Ingredients (kg)	Starter	Grower	Finisher
Corn	33.11	49.13	43.15
Broken rice	10.00	10.00	-
Wheat	10.00	-	20.00
Wheat bran	2.00	-	-
Full-fat soybean	10.00	10.00	12.00
Soybean meal 44%	31.46	27.63	20.16
Acid oil	-	0.25	2.15
Monodicalcium phosphate	0.95	0.85	0.52
Limestone	1.20	1.05	0.70
Sodium bicarbonate	0.20	0.20	0.30
Choline	0.03	0.03	0.00
DL-methionine	0.36	0.30	0.26
L-Lysine	0.18	0.13	0.14
L-threonine	0.10	0.07	0.06
Phytase	0.01	0.01	0.01
NSP Enzyme	0.01	0.01	0.01
Salt	0.25	0.20	0.20
Mineral Premix ^1^	0.10	0.10	0.10
Vitamin Premix ^2^	0.04	0.04	0.04
Total (kg)	100.00	100.00	100.00
** *Nutrients by calculation* **			
AME, Kcal/kg	3000.00	3100.00	3200.00
Crude protein, %	23.50	21.50	19.50
Fiber, %	3.10	2.92	3.05
Fat, %	4.13	4.65	6.95
Methionine, %	0.56	0.51	0.47
Methionine + Cystine, %	1.08	0.99	0.90
Lysine, %	1.44	1.29	1.15
Threonine, %	0.97	0.88	0.78
Valine, %	1.11	1.00	0.89
Isoleucine, %	0.97	0.89	0.80
Arginine, %	1.52	1.37	1.21
Tryptophan, %	0.23	0.21	0.18
Calcium, %	0.96	0.88	0.78
Total phosphorus, %	0.72	0.67	0.62
Available phosphorus, %	0.48	0.44	0.39
Na, %	0.23	0.23	0.20
DEB, mEq/kg	292.63	266.10	233.49

NSP: Non-starch polysaccharides; AME: Apparent Metabolizable Energy; ^1^ The mineral premix supplied the following per kg of complete feed: Cu, 16 mg; Zn, 120 mg; Fe, 20 mg; Mn, 120 mg; I, 1.25 mg; Selenium, 0.30 mg. ^2^ The vitamin premix supplied the following per kg of complete feed: vitamin A, 11,500 IU; vitamin K_3_, 3.60 mg; vitamin B_1_, 4 mg; vitamin B_2_, 8 mg; vitamin B_12_, 0.025 mg; vitamin E, 65 IU; biotin, 0.28 mg; folic acid, 2.00 mg; pantothenic acid, 25 mg; niacin, 65 mg (vitamin D_3_ not included). DEB: Dietary electrolyte balance, calculated as Na + K − Cl (mEq/kg diet).

**Table 2 animals-15-02900-t002:** Vitamin D_3_ and 25(OH)D_3_ supplementation in treatment diets (per kg).

Treatments	Starter	Grower	Finisher
NC	N	N	N
PC	Vit D_3_ 5000 IU	Vit D_3_ 4500 IU	Vit D_3_ 4000 IU
PC+25(OH)D_3__1X	Vit D_3_ 5000 IU +25(OH)D_3_ 1394 IU	Vit D_3_ 4500 IU +25(OH)D_3_ 1394 IU	Vit D_3_ 4000 IU +25(OH)D_3_ 1394 IU
PC+25(OH)D_3__2X	Vit D_3_ 5000 IU +25(OH)D_3_ 2788 IU	Vit D_3_ 4500 IU +25(OH)D_3_ 2788 IU	Vit D_3_ 4000 IU +25(OH)D_3_ 2788 IU

N = Not supplementing vitamin D_3_ in premix. The NC premix did not contain vitamin D_3_, so any vitamin D in this diet was provided only by the feed ingredients.

**Table 3 animals-15-02900-t003:** Effects of vitamin D_3_ and 25(OH)D_3_ supplementation on growth performance of broiler chickens.

Item	NC	PC	PC+	PC+	SEM	*p*-Value		
			25(OH)D_3__1X	25(OH)D_3__2X		ANOVA	L	Q
*Starter phase (0–11 days)*								
BW (g/b)	294.45 ^b^ ± 19.34	306.98 ^a^ ± 14.47	310.13 ^a^ ± 8.41	314.91 ^a^ ± 10.24	2.06	<0.01	<0.01	0.30
BWG (g/b)	251.55 ^b^ ± 19.34	264.09 ^a^ ± 14.47	267.23 ^a^ ± 8.41	272.01 ^a^ ± 10.24	2.06	<0.01	<0.01	0.30
ADG (g/b)	22.87 ^b^ ± 1.76	24.01 ^a^ ± 1.32	24.29 ^a^ ± 0.76	24.73 ^a^ ± 0.93	0.19	<0.01	<0.01	0.30
FI (g/b)	332.69 ± 15.19	348.03 ± 21.34	346.41 ± 14.82	346.12 ± 23.95	2.63	0.14	0.10	0.13
FCR	1.33 ± 0.06	1.32 ± 0.06	1.30 ± 0.04	1.27 ± 0.08	0.01	0.11	0.02	0.65
Mortality (%)	0.84 ± 2.14	0.42 ± 1.57	0.00 ± 0.00	0.42 ± 1.57	0.20	0.56	0.37	0.31
*Grower phase (12–30 days)*								
BW (g/b)	1719.05 ^b^ ± 50.49	1784.05 ^a^ ± 42.57	1764.88 ^a^ ± 64.56	1803.95 ^a^ ± 74.06	8.78	<0.01	<0.01	0.42
BWG (g/b)	1408.93 ^b^ ± 47.24	1469.14 ^a^ ± 42.05	1470.43 ^a^ ± 52.91	1496.96 ^a^ ± 66.70	8.15	<0.01	<0.01	0.24
ADG (g/b)	74.15 ^b^ ± 2.49	77.32 ^a^ ± 2.21	77.39 ^a^ ± 2.78	78.79 ^a^ ± 3.51	0.43	<0.01	<0.01	0.24
FI (g/b)	2120.73 ± 86.19	2168.44 ± 59.97	2116.09 ± 84.31	2151.33 ± 88.31	10.86	0.27	0.68	0.77
FCR	1.51 ^a^ ± 0.05	1.48 ^ab^ ± 0.05	1.44 ^b^ ± 0.05	1.44 ^b^ ± 0.07	0.01	<0.01	<0.01	0.36
Mortality (%)	3.78 ± 2.92	3.78 ± 4.95	4.20 ± 4.86	2.94 ± 4.47	0.57	0.89	0.69	0.59
*Finisher phase (31–40 days)*								
BW (g/b)	2793.03 ^b^ ± 122.26	2913.80 ^a^ ± 83.91	2870.30 ^a^ ± 72.81	2934.38 ^a^ ± 78.82	13.96	<0.01	<0.01	0.25
BWG (g/b)	1073.98 ± 103.45	1122.61 ± 63.48	1105.43 ± 56.11	1130.43 ± 79.72	10.53	0.24	0.11	0.57
ADG (g/b)	107.40 ± 10.35	112.26 ± 6.35	110.54 ± 5.61	113.04 ± 7.97	1.05	0.24	0.11	0.57
FI (g/b)	1864.01 ± 83.20	1863.97 ± 118.42	1900.74 ± 87.47	1894.34 ± 64.09	11.97	0.58	0.24	0.90
FCR	1.75 ± 0.14	1.66 ± 0.11	1.72 ± 0.11	1.68 ± 0.12	0.02	0.24	0.34	0.49
Mortality (%)	2.52 ± 3.80	2.10 ± 2.92	3.36 ± 5.01	2.10 ± 2.92	0.49	0.79	1.00	0.68
*Overall (0–40 days)*								
BW (g/b)	2793.03 ^b^ ± 122.26	2913.80 ^a^ ± 83.91	2870.30 ^a^ ± 72.81	2934.38 ^a^ ± 78.82	13.96	<0.01	<0.01	0.25
BWG (g/b)	2750.14 ^b^ ± 122.26	2870.91 ^a^ ± 83.91	2827.41 ^a^ ± 72.81	2891.48 ^a^ ± 78.82	13.96	<0.01	<0.01	0.25
ADG (g/b)	68.75 ^b^ ± 3.06	71.77 ^a^ ± 2.10	70.69 ^a^ ± 1.82	72.29 ^a^ ± 1.97	0.35	<0.01	<0.01	0.25
FI (g/b)	4317.42 ± 133.62	4380.43 ± 122.00	4363.24 ± 117.04	4391.79 ± 105.75	16.06	0.38	0.16	0.59
FCR	1.57 ^a^ ± 0.04	1.53 ^b^ ± 0.03	1.54 ^b^ ± 0.03	1.52 ^b^ ± 0.03	0.01	<0.01	<0.01	0.27
Mortality (%)	7.56 ± 6.29	5.58 ± 6.10	7.56 ± 6.70	5.46 ± 4.87	0.79	0.70	0.52	0.90

^ab^ Means with different superscripts within the same row are significantly different (*p* < 0.05). Values are presented as mean ± standard deviation (mean ± SD). SEM = standard error of means. Orthogonal contrasts: L = Linear; Q = Quadratic.

**Table 4 animals-15-02900-t004:** Effects of vitamin D_3_ and 25(OH)D_3_ supplementation on bone mineralization and strength of broiler chickens.

Item	NC	PC	PC+	PC+	SEM	*p*-Value		
			25(OH)D_3__1X	25(OH)D_3__2X		ANOVA	L	Q
Tibia ash (%)	41.30 ± 1.70	42.08 ± 2.40	42.44 ± 1.38	42.52 ± 1.51	0.20	0.51	0.08	0.49
Bone-breaking strength (N)	30.26 ± 5.23	32.03 ± 6.63	33.49 ± 4.87	35.86 ± 7.04	0.93	0.21	0.06	0.89

Data are presented as mean ± standard deviation (mean ± SD). SEM = standard error of means. Orthogonal contrasts: L = Linear, Q = Quadratic.

## Data Availability

The data presented in this study are available on request from the corresponding author upon reasonable request.

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
