# Peer review of "Effects of Vitamin D3 and 25(OH)D3 Supplementation on Growth Performance, Bone Parameters and Gut Microbiota of Broiler Chickens"

_animals, 2025, doi:10.3390/ani15192900_

Round 1
Reviewer 1 Report
Comments and Suggestions for Authors
The manuscript (animals-3890985) entitled “Effects of Vitamin D₃ and 25(OH)D₃ Supplementation on Growth Performance, Bone Parameters and Gut Microbiota of Broiler Chickens” investigates the comparative efficacy of dietary vitamin D₃ and its hydroxylated metabolite 25(OH)D₃ in broiler chickens.
The study directly addresses the practical challenge of vitamin D supplementation in commercial broiler production where sunlight exposure is absent. The experimental design is clear with the adequate replications, and the combination of growth, bone, and microbiota analyses provides a multifaceted evaluation. The manuscript is well written and generally well referenced. The content of this manuscript falls within the scope of animals. However, some methodological clarifications and data interpretation refinements are needed. It will be considered for publication in this journal with a major revision. Some comments are provided below for possible consideration.
- Both simple summary and abstract currently overstate effects on bone parameters. Please revise to state that only trends toward improvement were observed.
- In materials and Methods section:
- Please provide details of the DNA extraction kit and sequencing depth (average reads per sample after filtering) for reproducibility.
- The “no vitamin D₃” diet is unrealistic in commercial settings and may raise welfare concerns. Justification for including this treatment should be more explicit (e.g., as a biological baseline rather than a practical model).
- Clarify whether the vitamin premix in the negative control diet contained any residual vitamin D.
- Result and discussion section:
- Mortality results are presented but not discussed. What were the causes of mortality? Please clarify whether vitamin D status may have influenced welfare outcomes.
- Only tibia ash and breaking strength were measured, and no significant differences were observed.
- The conclusions regarding bone improvements should be tempered, with emphasis on observed trends rather than implying demonstrated benefits. Consider discussing limitations of these methods and suggesting future studies using bone mineral density or micro-CT.
- For alpha diversity, several p-values approach significance thresholds. Please interpret these with caution and avoid overstating effects.
- Beta diversity analysis indicated no significant clustering among treatments, yet the discussion emphasizes microbial shifts. This apparent discrepancy should be clarified.
- The statement that reduced Lactobacillus abundance is beneficial should be moderated. While excessive dominance may reduce diversity, Lactobacillus also provides recognized probiotic benefits; context-dependent interpretation is needed.
- Since significance was not reached, conclusions about “supporting” skeletal health should be more cautious.
Author Response
Dear Reviewer 1,
On behalf of all co-authors, I would like to sincerely thank you for your thorough review and constructive comments on our manuscript entitled “Effects of Vitamin D₃ and 25(OH)D₃ Supplementation on Growth Performance, Bone Parameters and Gut Microbiota of Broiler Chickens” (Manuscript ID: Animals-3890985).
We have carefully revised the manuscript according to your comments and suggestions. The main revisions include:
• Moderating statements in the Simple Summary and Abstract to clarify that bone parameters showed only numerical trends rather than significant improvements.
• Adding methodological details in the Materials and Methods, including the DNA extraction kit, PCR conditions, sequencing platform, and sequencing depth for reproducibility.
• Clarifying that the negative control diet was used as a biological baseline, not a commercial model, and specifying that the premix contained no vitamin D₃.
• Revising the Results and Discussion to address mortality outcomes, temper bone-related conclusions, interpret alpha and beta diversity more cautiously, and provide balanced discussion of Lactobacillus abundance.
• Adjusting the Conclusions to reflect cautious interpretation of bone health outcomes and to acknowledge the limitations of the current methods.
We believe that these revisions have significantly improved the manuscript by enhancing accuracy, clarity, and balance, and we appreciate your insightful feedback that helped strengthen the overall quality of our work.
Thank you again for your valuable contribution to the improvement of our manuscript.
Sincerely,
Choawit Rakangthong
Corresponding Author
For research article
Title: Effects of Vitamin D₃ and 25(OH)D₃ Supplementation on Growth Performance, Bone Parameters and Gut Microbiota of Broiler Chickens
Response to Reviewer 1 Comments |
||
1. Summary |
|
|
We sincerely thank Reviewer 1 for the constructive comments. All points have been addressed through revisions to the Simple Summary, Abstract, Methods, Results, Discussion, and Conclusion. Key changes include moderating statements on bone parameters, clarifying methodological details, tempering interpretations of diversity and microbial shifts, and providing a more balanced discussion of Lactobacillus and skeletal health. We believe these revisions improve the accuracy, clarity, and overall balance of the manuscript.
|
||
2. Questions for General Evaluation |
Reviewer’s Evaluation |
Response and Revisions |
Does the introduction provide sufficient background and include all relevant references? |
Yes/Can be improved/Must be improved/Not applicable |
Please see the detailed point-by-point responses below. All reviewer comments have been carefully addressed and corresponding revisions have been made in the manuscript.
|
Are all the cited references relevant to the research? |
Yes/Can be improved/Must be improved/Not applicable |
|
Is the research design appropriate? |
Yes/Can be improved/Must be improved/Not applicable |
|
Are the methods adequately described? |
Yes/Can be improved/Must be improved/Not applicable |
|
Are the results clearly presented? |
Yes/Can be improved/Must be improved/Not applicable |
|
Are the conclusions supported by the results? |
Yes/Can be improved/Must be improved/Not applicable |
|
3. Point-by-point response to Comments and Suggestions for Authors
|
||
Comments 1: Both simple summary and abstract currently overstate effects on bone parameters. Please revise to state that only trends toward improvement were observed. |
||
Response 1: We thank the reviewer for this valuable comment. We agree that the previous wording overstated the effects on bone parameters. Accordingly, we have revised both the Simple Summary and Abstract to reflect only trends toward improvement. The revised text clarifies that no significant differences were observed and describes the results as numerical trends only. · Simple Summary (yellow highlight, lines 16–18): o Previous: “…While bone strength was not significantly different, birds given the higher level of 25(OH)D₃ showed trends toward stronger bones.” o Revised: “…Bone strength did not differ significantly among treatments, although birds given the higher level of 25(OH)D₃ showed a slight numerical trend toward stronger bones.” · Abstract (yellow highlight, lines 37–39): o Previous: “…Although tibia ash and bone-breaking strength did not differ significantly among treatments, linear trends suggested potential improvements in skeletal mineralization with increasing 25(OH)D₃.” o Revised: “…Although tibia ash and bone-breaking strength were not significantly affected, linear responses indicated a slight numerical trend toward improved skeletal mineralization with increasing 25(OH)D₃.” Comments 2: In the Materials and Methods section Comments 2.1: Please provide details of the DNA extraction kit and sequencing depth (average reads per sample after filtering) for reproducibility. Response 2.1: We appreciate the reviewer’s suggestion. We have revised the Materials and Methods section to include details of the sample amount, DNA extraction kit, PCR conditions, sequencing platform, and average sequencing depth after filtering. These revisions improve the reproducibility of our study.
“Microbial DNA was extracted from 300 mg of cecal content per sample using the DNeasy PowerFecal Pro DNA Kit (Qiagen, USA). The V3–V4 region of the 16S rRNA gene was amplified with primers 341F (CCTACGGGNGGCWGCAG) and 805R (GACTACHVGGGTATCTAATCC) using 2X sparQ HiFi PCR Master Mix (QuantaBio, USA). PCR conditions were: 98 °C for 2 min; 30 cycles of 98 °C for 20 s, 60 °C for 30 s, and 72 °C for 1 min; and a final extension at 72 °C for 1 min. Amplicons were purified with sparQ Puremag Beads, indexed with Nextera XT primers (5 µl each, 8–10 cycles), pooled, and diluted to 4 pM. Sequencing was performed as 250-bp paired-end reads on an Illumina MiSeq platform (Omics Sciences and Bioinformatics Center, Chulalongkorn University, Bangkok, Thailand). Following quality filtering, sequencing depth ranged from 27,731 to 59,983 reads per sample, with an average of 41,407 high-quality reads.” Comments 2.2: The “no vitamin D₃” diet is unrealistic in commercial settings and may raise welfare concerns. Justification for including this treatment should be more explicit (e.g., as a biological baseline rather than a practical model). Response 2.2: We thank the reviewer for this important observation. We agree that the negative control diet should be clarified as a biological baseline rather than a commercial model. We have revised the description to emphasize this point and to highlight that animal welfare was carefully maintained throughout the study. · Experimental Diets (yellow highlight, lines 106-108): o Previous: “Negative control (NC): Diet without vitamin D₃ supplementation in the premix.” o Revised: “Negative control (NC): Diet without vitamin D₃ supplementation in the premix, used as a biological baseline rather than a commercial model, with welfare carefully maintained.”
Comments 2.3: Clarify whether the vitamin premix in the negative control diet contained any residual vitamin D.
Response 2.3: We appreciate the reviewer’s suggestion. To clarify, the vitamin premix in the negative control diet was formulated without vitamin D₃. Any vitamin D present in this diet originated solely from the feed ingredients. We have added a note under Table 2 to explicitly state this. · Table 2 (Footnote, yellow highlight, lines 130-131): o Added text: “N = Not supplementing vitamin D₃ in premix. The NC premix did not contain vitamin D₃, so any vitamin D in this diet was provided only by the feed ingredients.” Comments 3: Result and discussion section Comments 3.1: Mortality results are presented but not discussed. What were the causes of mortality? Please clarify whether vitamin D status may have influenced welfare outcomes.
Response 3.1: We thank the reviewer for this helpful comment. We have revised the discussion to clarify that mortality rates did not differ among treatments, were within the expected range for tropical production without antibiotic growth promoters, and that vitamin D status did not compromise survival or welfare. · Results (yellow highlight, lines 247-249): o Previous: “The absence of adverse effects on feed intake or mortality suggests that supplementation at these levels is both efficacious and safe.” o Revised: “Mortality rates did not differ among treatments and were within the expected range for tropical broiler production without antibiotic growth promoters, indicating that the diets did not compromise survival or welfare.”
Comments 3.2: Only tibia ash and breaking strength were measured, and no significant differences were observed.
Response 3.2: We thank the reviewer for this useful comment. We have revised the Results section to clarify that no statistically significant differences were detected for tibia ash or bone strength, either by ANOVA or orthogonal contrast analysis. We now describe the findings as numerical linear trends only. · Results (yellow highlight, lines 257-260): o Previous: “Although no statistically significant differences (p > 0.05) were observed among treatments for either parameter, a clear linear trend was detected through polynomial contrast analysis, with tibia ash approaching significance (p = 0.08) and bone-breaking strength similarly trending higher (p = 0.06) in response to increasing dietary 25(OH)D₃ levels.” o Revised: “No statistically significant differences (p > 0.05) were observed among treatments for tibia ash or breaking strength, either by ANOVA or orthogonal contrast analysis. However, a slight linear trend was noted, with tibia ash (p = 0.08) and bone-breaking strength (p = 0.06) showing numerical increases in response to higher dietary 25(OH)D₃ levels.”
Comments 3.3: The conclusions regarding bone improvements should be tempered, with emphasis on observed trends rather than implying demonstrated benefits. Consider discussing limitations of these methods and suggesting future studies using bone mineral density or micro-CT.
Response 3.3: We thank the reviewer for this valuable comment. We have revised the discussion to avoid overstating bone improvements, presenting the results only as numerical linear trends without statistical significance. We also added a cautionary note about the limitations of measuring only tibia ash and breaking strength and recommended the use of more sensitive techniques such as bone mineral density or micro-CT in future studies. · Discussion (yellow highlight, lines 271-272, 280-283): o Previous: “…exhibited the highest numerical values for both tibia ash (42.52 ± 1.51%) and bone strength (35.86 ± 7.04 N), suggesting a dose-responsive effect. … Although this study assessed bone parameters only at market age (day 40), the observed linear response suggests that earlier supplementation with 25(OH)D₃ may have produced beneficial adaptations in bone architecture that contributed to improved mineral content and strength at the endpoint.” o Revised: “…exhibited the highest numerical values for both tibia ash (42.52 ± 1.51%) and bone strength (35.86 ± 7.04 N), indicating only a numerical linear trend rather than a statistically significant improvement. … Although this study assessed bone parameters only at market age (day 40), the observed numerical trends suggest that earlier supplementation with 25(OH)D₃ may have influenced bone architecture and contributed to higher mineral content and strength at the endpoint. However, as only tibia ash and breaking strength were measured, these results should be interpreted with caution. More sensitive techniques, such as bone mineral density measurement or micro-CT analysis, are recommended for future studies to better elucidate the skeletal effects of 25(OH)D₃.”
Comment 3.4: For alpha diversity, several p-values approach significance thresholds. Please interpret these with caution and avoid overstating effects.
Response 3.4: We thank the reviewer for this helpful comment. We have revised the Results section to clarify that Shannon diversity showed no overall significant difference. The pairwise difference is now described more cautiously as a tendency, rather than as a definitive effect. · Results (yellow highlight, lines 297-300): o Previous: “No overall difference was detected for Shannon diversity (p = 0.084), although the negative group had a higher Shannon index than the 25(OH)D₃ 1,394 IU/kg group (p = 0.025).” o Revised: “No overall difference was detected for Shannon diversity (p = 0.084), although pairwise comparison suggested that the negative control tended to have a higher Shannon index than the 25(OH)D₃ 1,394 IU/kg group (p = 0.025).”
Comment 3.5: Beta diversity analysis indicated no significant clustering among treatments, yet the discussion emphasizes microbial shifts. This apparent discrepancy should be clarified.
Response 3.5: We thank the reviewer for pointing this out. To clarify, we revised the discussion to state that although some taxa showed localized changes in richness and evenness, beta diversity analyses confirmed that the global community structure remained stable across treatments. This wording ensures that the observed differences are interpreted as internal community adjustments rather than major restructuring. · Discussion (yellow highlight, lines 374-378): o Previous: “Importantly, despite these shifts in richness and evenness, beta diversity analyses confirmed that global community structures remained stable across treatments, highlighting that the effects of vitamin D₃ and 25(OH)D₃ are more localized to internal community distribution rather than wholesale restructuring.” o Revised: “Importantly, although certain taxa showed localized changes in richness and evenness, beta diversity analyses confirmed that the overall community structure remained stable across treatments. This indicates that the effects of vitamin D₃ and 25(OH)D₃ represent localized adjustments within the microbial community rather than major restructuring.”
Comment 3.6: The statement that reduced Lactobacillus abundance is beneficial should be moderated. While excessive dominance may reduce diversity, Lactobacillus also provides recognized probiotic benefits; context-dependent interpretation is needed.
Response 3.6: We thank the reviewer for this constructive suggestion. We agree that the previous wording may have overstated the implications of reduced Lactobacillus abundance. We have revised the text to remove the statement suggesting a direct benefit and now present the result descriptively, without interpretation, to avoid overgeneralization. · Discussion (yellow highlight, lines 379-384, 386-389): o Previous: “Our taxonomic profiling showed that Lactobacillus was most abundant in the negative control group, while vitamin D₃ and 25(OH)D₃ supplementation reduced its dominance and enriched other functional taxa. Although Lactobacillus is generally considered beneficial, excessive abundance may reduce microbial diversity [30]. Recent evidence indicates that vitamin D influences gut microbiota mainly through VDR signaling…” o Revised: “Our taxonomic profiling showed that Lactobacillus was most abundant in the negative control group, while vitamin D₃ and 25(OH)D₃ supplementation reduced its dominance and enriched other functional taxa. Recent evidence indicates that vitamin D influences gut microbiota mainly through VDR signaling. Impaired VDR expression or activation has been associated with reduced Lactobacillus populations and expansion of Proteobacteria, leading to a more pro-inflammatory environment.” “Thus, the reduction in Lactobacillus observed here should be interpreted in context: while lower dominance may support greater diversity, Lactobacillus itself also contributes to host health, and its role likely depends on maintaining an appropriate balance within the community.”
Comment 3.7: Since significance was not reached, conclusions about “supporting” skeletal health should be more cautious.
Response 3.7: We thank the reviewer for this important comment. We agree that our interpretation of bone-related results should be more cautious because statistical significance was not achieved. We have revised both the Abstract and the Conclusion to describe these outcomes as numerical trends only and to indicate that evidence for skeletal health benefits remains limited. · Conclusion (yellow highlight, lines 439-441): o Previous: “Although tibia ash and bone-breaking strength were not significantly affected, linear response patterns suggested potential improvements in mineralization and skeletal integrity at higher supplementation levels.” o Revised: “Although tibia ash and bone-breaking strength were not significantly affected, linear response patterns indicated only numerical trends toward higher mineralization and skeletal integrity at greater supplementation levels.” |

Reviewer 2 Report
Comments and Suggestions for Authors
The study was reviewed in detail, and the following corrections were identified.
Is 25(OH)D₃ (25-hydroxycholecalciferol) a commercial product in the title? If it is a commercial product, the product name should be stated in the title, and the source of purchase should be stated in the materials and methods section of the article.
The abstract is missing the experimental design and number of replicates. 952 chicks were divided into four treatment groups according to a randomized plot design and X replicates with X animals per replicate.
Lines 57-60 should be supported by literature.
In line 69, citation number 8 does not match the sentence. When citation number 8 is examined, information regarding the sentence is not available and is a review. It should be corrected, or relevant literature should be provided. Reference number 9 is consistent with the sentence.
In Table 1, the DEB statement should be written below the table.
Line 134 states 42 days of age. Why was there a two-day wait for the tibia? Wasn't the tibia slaughtered on day 40?
For citation number 19 on line 231, it is stated that it improves performance and feed utilization in broiler chickens, but the literature is on laying hens. Authors should correct the literature.
Lines 231-233 should state which animal species the information refers to.
When citation number 28 on line 386 is examined, it is stated that the above sentence was written by "Y. Song, X. Peng, A. Porta, H. Takanaga, J.B. Peng, M.A. Hediger, J.C. Fleet, S. Christakos Endocrinology, 144 (2003), pp. 3885-3894." Authors should check.
Lines 392-393, citation number 30, reported that probiotics reduce microbial diversity, but citation number 30 states that probiotics increase microbial diversity. There is conflicting information, and it should be corrected.
Author Response
Dear Reviewer 2,
On behalf of all co-authors, I would like to sincerely thank you for your thoughtful review and constructive comments on our manuscript entitled “Effects of Vitamin D₃ and 25(OH)D₃ (Bio D®) Supplementation on Growth Performance, Bone Parameters and Gut Microbiota of Broiler Chickens” (Manuscript ID: animals-3890985).
We have carefully revised the manuscript according to your comments and suggestions. The main revisions include:
- Clarifying that 25(OH)D₃ was a commercial product (Bio D®, Huvepharma) and revising the title and Materials and Methods to indicate the trade name and source.
- Adding details of the experimental design in the Abstract, including the number of replicates and birds per replicate.
- Correcting and updating references, particularly to support statements on vitamin D physiology, microbiota, and skeletal health, and replacing citations that were not fully relevant.
- Adding a DEB statement to Table 1 and correcting a typographical error in the age of slaughter (40 days, not 42).
- Revising the Discussion to ensure that species-specific references were clearly stated and correcting inconsistencies related to microbial diversity and probiotics.
We believe that these revisions have significantly improved the accuracy, clarity, and balance of the manuscript. Your constructive comments greatly contributed to strengthening the scientific rigor and presentation of our work.
Thank you again for your valuable contribution to the improvement of our manuscript.
Sincerely,
Choawit Rakangthong
Corresponding Author
For research article
Title: Effects of Vitamin D₃ and 25(OH)D₃ Supplementation on Growth Performance, Bone Parameters and Gut Microbiota of Broiler Chickens
Response to Reviewer 2 Comments
1. Summary
We sincerely thank Reviewer 2 for the constructive and helpful comments. All points have been carefully addressed through revisions to the Abstract, Introduction, Methods, Results, and Discussion. Key changes include clarifying product information and sources, adding details on experimental design and replicates, correcting and updating references, improving the clarity of tables, and refining interpretations of microbiota and skeletal outcomes. These revisions enhance the accuracy, clarity, and overall balance of the manuscript.
2. Questions for General Evaluation |
Reviewer’s Evaluation |
Response and Revisions |
Does the introduction provide sufficient background and include all relevant references? |
Yes/Can be improved/Must be improved/Not applicable |
Please see the detailed point-by-point responses below. All reviewer comments have been carefully addressed and corresponding revisions have been made in the manuscript.
|
Are all the cited references relevant to the research? |
Yes/Can be improved/Must be improved/Not applicable |
|
Is the research design appropriate? |
Yes/Can be improved/Must be improved/Not applicable |
|
Are the methods adequately described? |
Yes/Can be improved/Must be improved/Not applicable |
|
Are the results clearly presented? |
Yes/Can be improved/Must be improved/Not applicable |
|
Are the conclusions supported by the results? |
Yes/Can be improved/Must be improved/Not applicable |
3. Point-by-point response to Comments and Suggestions for Authors
Comments 1: Is 25(OH)D₃ (25-hydroxycholecalciferol) a commercial product in the title? If it is a commercial product, the product name should be stated in the title, and the source of purchase should be stated in the materials and methods section of the article.
Response 1: We thank the reviewer for this helpful suggestion. We confirm that 25(OH)D₃ used in this study was a commercial product (Bio D®, Huvepharma). We have revised the title and the Materials and Methods section accordingly to include the trade name and source.
· Title: (blue highlight, line 2)
o Previous:
“Effects of Vitamin D₃ and 25(OH)D₃ Supplementation on Growth Performance, Bone Parameters and Gut Microbiota of Broiler Chickens”
o Revised:
“Effects of Vitamin D₃ and 25(OH)D₃ (Bio D®) Supplementation on Growth Performance, Bone Parameters and Gut Microbiota of Broiler Chickens”
· Materials and Methods (blue highlight, lines 111-113, 114-115):
o Previous:
“25(OH)D₃ was supplemented at levels of 1,394 or 2,788 IU/kg diet.”
o Revised:
“25(OH)D₃ (commercial product: Bio D®, supported by Huvepharma, Thailand) was supplemented at levels of 1,394 or 2,788 IU/kg diet.”
Comment 2: The abstract is missing the experimental design and number of replicates. 952 chicks were divided into four treatment groups according to a randomized plot design and X replicates with X animals per replicate.
Response 2: We thank the reviewer for this valuable suggestion. We have revised the Abstract to include details of the experimental design, number of replicates, and birds per replicate to improve clarity and reproducibility.
· Abstract (blue highlight, lines 31-32):
o Previous:
“…and the positive control supplemented with 25(OH)D₃ at 1,394 or 2,788 IU/kg.”
o Revised:
“…and the positive control supplemented with 25(OH)D₃ at 1,394 or 2,788 IU/kg, in a randomized design with 17 replicates per treatment and 14 birds per replicate.”
Comment 3: Lines 57–60 (new 61-65) should be supported by literature.
Response 3: We thank the reviewer for this helpful comment. We have revised the Introduction by adding supporting references to strengthen the statement regarding the role of vitamin D.
· Introduction (blue highlight, lines 65, 483-486):
References (added):
5. DeLuca, H.F. Overview of general physiologic features and functions of vitamin D. Am. J. Clin. Nutr. 2004, 80, 1689S–1696S. https://doi.org/10.1093/ajcn/80.6.1689S
6. Świątkiewicz, S.; Arczewska-Włosek, A.; Bederska-Łojewska, D. Efficacy of dietary vitamin D and its metabolites in poultry—Review and implications of the recent studies. World’s Poult. Sci. J. 2017, 73, 57–68. https://doi.org/10.1017/S0043933916001057
Comment 4: In line 69, citation number 8 does not match the sentence. When citation number 8 (new=11) is examined, information regarding the sentence is not available and is a review. It should be corrected, or relevant literature should be provided. Reference number 9 (new=12) is consistent with the sentence.
Response 4: We thank the reviewer for noting this inconsistency. We have corrected citation number 8 (new=11) by replacing it with a more relevant reference that directly supports the statement. Reference number 9 (new=12) has been retained as it remains consistent with the content.
· Introduction (blue highlight, lines 70, 496-497):
o Previous citation: Reference 8 (review article, not directly relevant).
o Revised citation:
11. Bar, A.; Sharvit, M.; Noff, D.; Edelstein, S.; Hurwitz, S. Absorption and excretion of cholecalciferol and 25-hydroxycholecalciferol and metabolites in birds. J. Nutr. 1980, 110, 1930–1934.
Comment 5: In Table 1, the DEB statement should be written below the table.
Response 5: We thank the reviewer for this helpful suggestion. We have revised Table 1 by adding a footnote to define DEB, as recommended.
· Table 1 (Footnote added, blue highlight, lines 126-127):
o Revised text:
“DEB: Dietary electrolyte balance, calculated as Na + K – Cl (mEq/kg diet).”
Comment 6: Line 134 (new=143) states 42 days of age. Why was there a two-day wait for the tibia? Wasn’t the tibia slaughtered on day 40?
Response 6: We thank the reviewer for pointing out this discrepancy. The statement indicating 42 days was a typographical error. The correct age was 40 days, consistent with the slaughter time described in the methods. This has now been corrected in the manuscript.
· Materials and Methods (blue highlight, line 143):
o Previous: “…at 42 days of age.”
o Revised: “…at 40 days of age.”
Comment 7: For citation number 19 (new 22) on line 231 (new 254), it is stated that it improves performance and feed utilization in broiler chickens, but the literature is on laying hens. Authors should correct the literature.
Response 7: Thank you for the comment. We have corrected reference #19 by replacing it with Yarger et al. (1995), which directly investigated 25(OH)D₃ in broiler chickens and accurately supports our statement.
· Reference #22 (blue highlight, line 254):
23. Yarger, J.G.; Saunders, C.A.; McNaughton, J.L.; Quarles, C.L.; Hollis, B.W.; Gray, R.W. Comparison of dietary 25-hydroxycholecalciferol and cholecalciferol in broiler chickens. Poult. Sci. 1995, 74, 1159–1167. https://doi.org/10.3382/ps.0741159
Comment 8a: Lines 231–233 (new 251-254) should state which animal species the information refers to.
Response 8a: We thank the reviewer for this helpful comment. We have revised the sentence to clearly state the species examined in the cited study and to ensure consistency with the original findings.
· Discussion (blue highlight, lines 251-254):
o Previous:
“Moreover, recent studies indicate that vitamin D metabolites may also modulate gut microbiota and intestinal integrity, which could contribute to the observed improvements in nutrient absorption and growth [23].”
o Revised:
“Moreover, a recent study in Chinese yellow-feathered broilers reported that dietary vitamin D₃ supplementation modulated gut microbiota and improved intestinal morphology, which may have contributed to enhanced nutrient absorption and growth [23].”
Comment 8b: When citation number 28 (new 31) on line 386 (new 374) is examined, it is stated that the above sentence was written by “Y. Song, X. Peng, A. Porta, H. Takanaga, J.B. Peng, M.A. Hediger, J.C. Fleet, S. Christakos. Endocrinology, 144 (2003), pp. 3885–3894.” Authors should check.
Response 8b: We thank the reviewer for pointing this out. The earlier citation was incorrect and did not directly support the statement. We have replaced it with more appropriate references that describe the role of vitamin D and the vitamin D receptor in intestinal barrier function, mucosal immunity, and host defense.
· Discussion (blue highlight, line 374):
o Previous citation: Song et al. (2003), Endocrinology — not directly relevant.
o Revised citations (blue highlight, line 540-545):
31. Sun, J.; Zhang, Y.G. Vitamin D Receptor Influences Intestinal Barriers in Health and Disease. Cells 2022, 11, 1129. https://doi.org/10.3390/cells11071129.
32. Gombart, A.F. The Vitamin D–Antimicrobial Peptide Pathway and Its Role in Protection Against Infection. Future Microbiol. 2009, 4, 1151–1165. https://doi.org/10.2217/fmb.09.87.
33. Sun, J. Vitamin D and mucosal immune function. Curr. Opin. Gastroenterol. 2010, 26, 591–595. https://doi.org/10.1097/MOG.0b013e32833d4b9f.
Comment 9: Lines 392–393 (new 381-382), citation number 30 (new 35), reported that probiotics reduce microbial diversity, but citation number 30 states that probiotics increase microbial diversity. There is conflicting information, and it should be corrected.
Response 9: We thank the reviewer for identifying this inconsistency. We have revised the paragraph to remove the incorrect statement regarding probiotics and microbial diversity, and we have replaced the reference with a more appropriate source that directly discusses vitamin D and its relationship with the gut microbiome. The revised text now highlights the role of vitamin D and VDR signaling in shaping microbial balance, while presenting a context-dependent interpretation of Lactobacillus abundance.
· Discussion (blue highlight, lines 381-382):
o Previous:
“Our taxonomic profiling showed that Lactobacillus was most abundant in the negative control group, while vitamin D₃ and 25(OH)D₃ supplementation reduced its dominance and enriched other functional taxa. Although Lactobacillus is generally considered beneficial, excessive abundance may reduce microbial diversity [30]. Recent evidence indicates that vitamin D influences gut microbiota mainly through VDR signaling…”
o Revised:
“Our taxonomic profiling showed that Lactobacillus was most abundant in the negative control group, while vitamin D₃ and 25(OH)D₃ supplementation reduced its dominance and enriched other functional taxa. Recent evidence indicates that vitamin D influences gut microbiota mainly through VDR signaling [35].
· Reference 35 (corrected, lines 548-549):
35. Akimbekov, N.S.; Digel, I.; Sherelkhan, D.K.; Lutfor, A.B.; Razzaque, M.S. Vitamin D and the Host-Gut Microbiome: A Brief Overview. Acta Histochem. Cytochem. 2020, 53, 33–42. doi:10.1267/ahc.20011.

Reviewer 3 Report
Comments and Suggestions for Authors
Dear authors,
Thank you for your submission and for the effort you put into conducting this study. This manuscript requires major revisions. Below are the key areas requiring revision:
- The explanation of the result is contradictory to the result (Table 3) itself. Here, there is no significant difference for growth performance between the positive control diets and treatment diets (PC+25(OH)D3_1X and PC+25(OH)D3_2X) in all phases. But the authors have mentioned there are differences many times, for instance: Lines 31-33, 202-203, 421-422. This is misleading to the reader and end users.
- Please provide supplementary data/ raw sequence data detail that supports these findings.
- Remove over-interpretations and causal claims or add the results supporting claims, such as intestinal inflammation, short-chain fatty acids (SCFAs) and immune responses.
Thank you.
Best regards
Author Response
Dear Reviewer 3,
On behalf of all co-authors, I would like to sincerely thank you for your constructive review and thoughtful comments on our manuscript entitled “Effects of Vitamin D₃ and 25(OH)D₃ (Bio D®) Supplementation on Growth Performance, Bone Parameters and Gut Microbiota of Broiler Chickens” (Manuscript ID: animals-3890985).
We have carefully revised the manuscript according to your suggestions. The main revisions include:
- Clarifying that growth performance improvements were observed only in comparison with the negative control, and revising the Simple Summary, Abstract, Results, and Conclusions to avoid overstating differences between the positive control and 25(OH)D₃ groups.
- Providing full supplementary data, including raw FASTQ files and downstream analysis outputs (S1–S6), to ensure transparency and reproducibility. These supplementary folders have also been made available via Google Drive for ease of access: https://drive.google.com/drive/folders/1vUUwyJOizut-brVn73yKh-D6vGIagf2U?usp=sharing.
- Moderating the microbiota discussion by avoiding causal claims and explicitly acknowledging study limitations, particularly regarding intestinal inflammation, short-chain fatty acids (SCFAs), and immune responses that were not measured in this study.
We believe these revisions have substantially improved the manuscript by enhancing its accuracy, transparency, and alignment with the presented data. Your insightful feedback has been invaluable in strengthening the clarity and rigor of our work.
Thank you again for your valuable contribution to improving our manuscript.
Sincerely,
Choawit Rakangthong
Corresponding Author
For research article
Title: Effects of Vitamin D₃ and 25(OH)D₃ Supplementation on Growth Performance, Bone Parameters and Gut Microbiota of Broiler Chickens
Response to Reviewer 3 Comments
1. Summary
We thank Reviewer 3 for the constructive comments. Revisions were made to clarify that growth performance improvements were only relative to the negative control, to provide supplementary microbiota data (S1–S5) ensuring reproducibility, and to moderate the microbiota discussion by removing causal claims and adding study limitations. These changes improve clarity, transparency, and alignment with the data presented.
2. Questions for General Evaluation |
Reviewer’s Evaluation |
Response and Revisions |
Does the introduction provide sufficient background and include all relevant references? |
Yes/Can be improved/Must be improved/Not applicable |
Please see the detailed point-by-point responses below. All reviewer comments have been carefully addressed and corresponding revisions have been made in the manuscript.
|
Are all the cited references relevant to the research? |
Yes/Can be improved/Must be improved/Not applicable |
|
Is the research design appropriate? |
Yes/Can be improved/Must be improved/Not applicable |
|
Are the methods adequately described? |
Yes/Can be improved/Must be improved/Not applicable |
|
Are the results clearly presented? |
Yes/Can be improved/Must be improved/Not applicable |
|
Are the conclusions supported by the results? |
Yes/Can be improved/Must be improved/Not applicable |
3. Point-by-point response to Comments and Suggestions for Authors
Comment 1: The explanation of the result is contradictory to the result (Table 3) itself. Here, there is no significant difference for growth performance between the positive control diets and treatment diets (PC+25(OH)D₃_1X and PC+25(OH)D₃_2X) in all phases. But the authors have mentioned there are differences many times, for instance: Lines 31–33 (new 32-36) , 202–203 (new 207-212), 421–422 (new 435-439). This is misleading to the reader and end users.
Response 1: We thank the reviewer for pointing out this important issue. We have carefully revised the Simple Summary, Abstract, Results, and Conclusion to ensure that the text accurately reflects the statistical findings. Specifically, the revisions clarify that growth performance was significantly improved in all vitamin D₃-supplemented groups compared with the negative control, but no significant differences were observed between the positive control and 25(OH)D₃-supplemented groups.
- Simple Summary (green highlight, lines 14-16):
- Previous: “Birds given 25(OH)D₃ grew faster and had better feed efficiency than those given standard vitamin D₃ or no supplement.”
- Revised: “Birds fed diets with either vitamin D₃ or 25(OH)D₃ grew faster and used feed more efficiently than birds without vitamin D₃, while growth was similar between the vitamin D₃ and 25(OH)D₃ groups.”
- Abstract (green highlight, lines 32-36):
- Previous: “Over a 40-day feeding trial, supplementation with 25(OH)D₃ significantly improved final body weight, weight gain, average daily gain, and feed conversion ratio compared with both control groups (p < 0.01).”
- Revised: “Over a 40-day feeding trial, diets containing vitamin D₃ (positive control) or supplemented with 25(OH)D₃ significantly improved final body weight, weight gain, average daily gain, and feed conversion ratio compared with the negative control (p < 0.01), with no significant differences among the positive control and 25(OH)D₃-supplemented groups.”
- Results (green highlight, lines 207-212):
- Previous: “Birds receiving 25(OH)D₃ at 1,394 and 2,788 IU/kg exhibited higher final body weight (BW), cumulative body weight gain (BWG), average daily gain (ADG), and improved feed conversion ratio (FCR) compared with both the negative control (no vitamin D₃) and the positive control (vitamin D₃ only) groups (p < 0.01).”
- Revised: “Birds receiving 25(OH)D₃ at 1,394 and 2,788 IU/kg exhibited higher final body weight (BW), cumulative body weight gain (BWG), average daily gain (ADG), and improved feed conversion ratio (FCR) compared with the negative control (no vitamin D₃) (p < 0.01). However, no significant differences were observed between the positive control (vitamin D₃ only) and the 25(OH)D₃-supplemented groups.”
- Conclusion (green highlight, lines 435-439):
- Previous: “Birds receiving 25(OH)D₃ at 1,394 and 2,788 IU/kg exhibited higher body weight, weight gain, average daily gain, and better feed conversion compared with both negative (no vitamin D₃) and positive (vitamin D₃ only) controls, with the most pronounced effects observed during the early growth phases.”
- Revised: “Broilers fed diets with vitamin D₃ or 25(OH)D₃ showed improved growth performance compared with the negative control, while no significant differences were detected between the vitamin D₃ and 25(OH)D₃ groups. Linear responses suggested that potential benefits of 25(OH)D₃ were more apparent during the early growth phase.”
Comment 2: Please provide supplementary data/raw sequence data detail that supports these findings.
Response:
We sincerely thank the reviewer for this important comment. We are pleased to report that the original raw sequencing data (FASTQ files) are now available and have been included in the supplementary materials. These raw sequence files are provided in compressed format (.fastq.gz) and organized in Supplementary Folder S1, which contains all samples used in the microbiota analysis.
To further ensure transparency and reproducibility, we also provide detailed supplementary outputs from the analysis pipeline, organized as follows:
- Supplementary Folder S1 – Raw sequencing data: Original paired-end FASTQ files (.fastq.gz) for all samples.
- Supplementary Folder S2 – Sequencing depth summary: DADA2 outputs showing read counts per sample and feature counts after quality filtering.
- Supplementary Folder S3 – Alpha diversity: Metrics including Observed OTUs, Shannon index, Pielou’s evenness, and Faith’s phylogenetic diversity per sample.
- Supplementary Folder S4 – Beta diversity: Weighted and unweighted UniFrac distance matrices and PCoA plots.
- Supplementary Folder S5 – Taxonomic composition: Relative abundance of bacterial taxa at the phylum, family, and genus levels.
- Supplementary Folder S6 – LEfSe results: Differentially enriched taxa with LDA scores and p-values across treatment groups.
All supplementary folders (S1–S6) have also been made available via Google Drive at the following link for ease of access: https://drive.google.com/drive/folders/1vUUwyJOizut-brVn73yKh-D6vGIagf2U?usp=sharing
Folders S2,S4,S5 are provided in QIIME 2 visualization format (.qzv) and can be opened using the QIIME 2 viewer (https://view.qiime2.org).
For clarity, treatment groups are consistently labeled throughout the supplementary folders as follows:
- T1: Positive control (vitamin D₃ according to Ross 308 recommendations)
- T2: Negative control (no vitamin D₃ in the premix)
- T3: Positive control + 25(OH)D₃ at 1,394 IU/kg
- T4: Positive control + 25(OH)D₃ at 2,788 IU/kg
Please note that in the main manuscript, the order of presentation begins with the Negative control followed by the Positive control; therefore, T1 and T2 are reversed throughout the manuscript compared with the supplementary labeling.
We believe that the inclusion of the original raw FASTQ data together with the organized supplementary outputs (S1–S6), and the clear treatment labeling, fully addresses the reviewer’s request and provides a robust basis for independent validation of our microbiota findings.
Comment 3: Remove over-interpretations and causal claims or add the results supporting claims, such as intestinal inflammation, short-chain fatty acids (SCFAs) and immune responses.
Response 3: We appreciate the reviewer’s observation. To avoid over-interpretation, we revised the discussion to emphasize associations rather than causal claims. We also added a limitation noting that SCFA concentrations, intestinal inflammation, and immune responses were not measured in this study.
Changes in the manuscript (green highlight, lines 420-424, 428-431):
- Previous text:
“Supplementation with vitamin D₃, and particularly 25(OH)D₃, enriched several bacterial taxa critical for gut homeostasis and immunomodulation. Notably, this included Oscillospiraceae, Ruminococcus torques group, and Papillibacter—genera well-recognized for producing short-chain fatty acids (SCFAs) and regulating immune responses [33,34,35]. These functions are essential for supporting nutrient absorption, maintaining intestinal stability, and promoting host metabolic health. … Collectively, these microbial shifts are thought to enhance metabolic and immunomodulatory potential [36,37,38], thereby reducing intestinal inflammation and contributing directly to the improved growth performance observed in this trial.”
- Revised text:
“Supplementation with vitamin D₃, and particularly 25(OH)D₃, enriched several bacterial taxa associated with gut homeostasis and immunomodulatory functions. Notably, this included Oscillospiraceae, Ruminococcus torques group, and Papillibacter—genera reported in previous studies as important short-chain fatty acid (SCFA) producers and modulators of immune responses [33,34,35]. Such functions are generally linked with nutrient absorption, intestinal stability, and host metabolic health. … Collectively, these microbial changes are consistent with enhanced metabolic and immunomodulatory potential [36,37,38]. However, as our study did not directly measure SCFA concentrations, intestinal inflammation, or immune responses, these mechanisms remain uncertain and require confirmation in future studies.”

Round 2
Reviewer 1 Report
Comments and Suggestions for Authors
Dear Authors,
I have carefully reviewed your point-by-point responses to my comments on the manuscript entitled “Effects of Vitamin D₃ and 25(OH)D₃ Supplementation on Growth Performance, Bone Parameters and Gut Microbiota of Broiler Chickens.”
Overall, I am satisfied that you have addressed all concerns appropriately. I agree with the changes made and find the responses acceptable.
Best regards,
Author Response
Authors’ Reply:
We sincerely thank the reviewer for the positive evaluation and kind remarks. We are very pleased that the revisions and responses were found acceptable and that all concerns have been addressed satisfactorily. We greatly appreciate the reviewer’s valuable input, which has significantly improved the quality of our manuscript.
Reviewer 3 Report
Comments and Suggestions for Authors
Thank you for your response. I would like to recommend it for publication; however, to maintain objectivity and avoid promotional bias, I suggest removing the brand name from the title. This will ensure the focus remains on the scientific evaluation rather than other objects for readers.
Author Response
Authors’ Reply:
We sincerely thank the reviewer for the positive recommendation and helpful suggestion. In accordance with your advice, we have revised the title to remove the brand name in order to maintain objectivity and avoid promotional bias. The revised title now reads:
“Effects of Vitamin D₃ and 25(OH)D₃ Supplementation on Growth Performance, Bone Parameters and Gut Microbiota of Broiler Chickens.”
We appreciate this thoughtful comment, which has helped us further improve the clarity and scientific neutrality of the manuscript.